# Detection of eye contact with deep neural networks is as accurate as human experts

Eunji Chong [1✉], Elysha Clark-Whitney[2], Audrey Southerland[1], Elizabeth Stubbs[1], Chanel Miller[1], Eliana L. Ajodan[2], Melanie R. Silverman[2], Catherine Lord[3], Agata Rozga [1], Rebecca M. Jones[2] & James M. Rehg[1]

Eye contact is among the most primary means of social communication used by humans. Quantification of eye contact is valuable as a part of the analysis of social roles and communication skills, and for clinical screening. Estimating a subject's looking direction is a challenging task, but eye contact can be effectively captured by a wearable point-of-view camera which provides a unique viewpoint. While moments of eye contact from this viewpoint can be hand-coded, such a process tends to be laborious and subjective. In this work, we develop a deep neural network model to automatically detect eye contact in egocentric video. It is the first to achieve accuracy equivalent to that of human experts. We train a deep convolutional network using a dataset of 4,339,879 annotated images, consisting of 103 subjects with diverse demographic backgrounds. 57 subjects have a diagnosis of Autism Spectrum Disorder. The network achieves overall precision of 0.936 and recall of 0.943 on 18 validation subjects, and its performance is on par with 10 trained human coders with a mean precision 0.918 and recall 0.946. Our method will be instrumental in gaze behavior analysis by serving as a scalable, objective, and accessible tool for clinicians and researchers.

[1] School of Interactive Computing, Georgia Institute of Technology, Atlanta, USA. [2] Center for Autism and the Developing Brain, Weill Cornell Medicine, New York, USA. [3] School of Medicine, University of California, Los Angeles, USA. ✉email: eunjichong@gatech.edu

Gaze behavior is a key foundation of face-to-face social interaction. Eye contact, the act of looking another person in the eyes, is one of the earliest social skills to emerge in development[1,2], and studies have shown that infants are tuned to looking at faces from birth[3,4]. Eye contact serves multiple important functions in social communication, including the establishment and recognition of relationships between partners and the expression of interest and attentiveness[5,6]. Moreover, it is a core component of joint attention, in coordination with other gestures[7], which is an important developmental milestone. Atypical use of eye contact and abnormal gaze patterns are often part of a list of red flags for numerous medical and/or psychiatric conditions, including autism spectrum disorder (ASD)[8,9], Fragile X syndrome[10], ADHD[11], Williams Syndrome[12], social anxiety/behavioral inhibition[13], and oppositional defiant disorder[14]. In particular, decreased eye contact is included in the DSM-5 diagnostic criteria for ASD[15], and is also a focus of early screening and treatment.

As a result of the critical importance of gaze, a variety of technologies have been developed to automate the measurement of gaze behavior, of which eye tracking is the best known example. Conventional monitor-based eye tracking is unsuitable for measuring the contingent real-world aspects of social gaze during face-to-face interactions. While wearable eye trackers can be utilized to measure gaze behavior in adults[16–18] and infants[19,20], they are both expensive and burdensome to the subject. The need to wear and calibrate eye tracking hardware can be a tremendous challenge to subjects with compliance, distraction or fatigue issues, and this can affect both the yield and quality of the data. Infants, young children, and individuals with health problems are examples of subject groups that are likely to have such difficulties. Moreover, since the eye tracker only provides the point of gaze in a captured video recording, manual region of interest annotation must be performed on the video in order to identify the gaze targets, limiting the scalability of the approach.

We have pioneered a novel, scalable, low-burden approach to automatically detecting moments of eye contact between individuals during face-to-face interactions[21], illustrated in Fig. 1. The interactive partner wears a low-cost pair of glasses with a point-of-view (PoV) camera embedded in the bridge, which serves as a video recorder. By virtue of its placement, the subject will be looking directly toward the camera any time they are making eye contact with the interactive partner, facilitating automatic detection of those looks toward the camera using computer vision methods. In our approach, the subject is completely unencumbered and the burden on the interactive partner is low since the glasses are light-weight and unobtrusive. Note that in our experiments we remove the lenses of the glasses to provide the subject with an unobstructed view of the interactive partner's eyes.

While human raters can achieve levels of agreement above 90% when identifying instances of eye contact in PoV videos[22,23], the accuracy of automated detection approaches achieved in prior works[21,24] is well below this level of performance, making automatic coding unusable by researchers and practitioners as a measurement tool. This paper addresses this challenge by exploring three directions. First, we hypothesize that modern deep learning architectures can leverage a large dataset of 4.7M human-annotated eye contact events to achieve higher accuracy. However, while our dataset is large by any standard, it only contains around 100 unique subjects. In contrast, datasets for face detection, recognition, and other tasks, which have been shown to yield high performance when using deep models[25–27], contain orders of magnitude more variability. Therefore, our second hypothesis is that we can close this gap by using task transfer learning from additional datasets that model the relationship between head pose and eye gaze direction, which is central to our task. Transfer learning is based on leveraging representations learned for one task to improve performance on a related task[28]. Third, we hypothesize that the frequency and duration of moments of eye contact identified by our automated method will

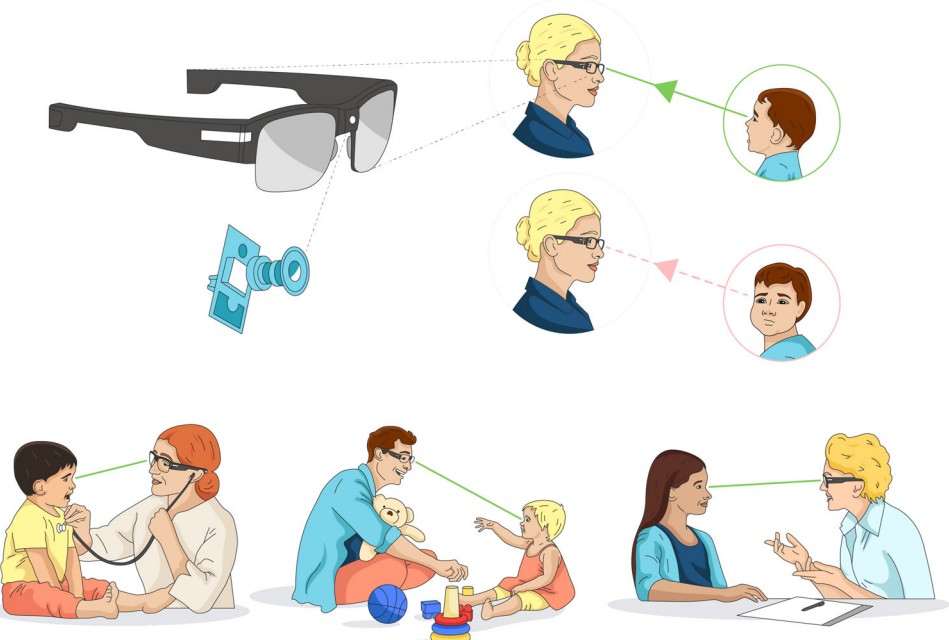

**Fig. 1 Overview of the approach.** Wearable glasses with a small outward-facing camera embedded in the bridge are used to record the face of the camera wearer's interactive partner. By virtue of its placement, gaze during eye contact is directed toward the camera, and is captured in video, enabling automated detection. Due to its ease of use, the approach can be widely deployed in a variety of settings, as illustrated in the figure, for which eye tracking may be infeasible due to cost, burden, compliance, or distraction issues.

correlate with measures of social impairment among individuals with ASD. Establishing this hypothesis validates the feasibility of fully automated eye contact coding using our approach.

## Results

**Representativeness of validation set**. In $t$-tests and $\chi^2$ tests that were run to confirm that subjects included in the validation set are representative of the overall sample, the validation set did not differ from the rest of the sample in terms of diagnostic group ($\chi = 0.09$, $p = 0.77$), gender ($\chi = 3.62$, $p = 0.06$), age ($t = 0.49$, $p = 0.62$, $M = 37.72$ vs. 36.17, $SD = 13.10$ vs. 12.15), race ($\chi = 2.70$, $p = 0.61$), ethnicity ($\chi = 0.29$, $p = 0.86$), or severity of social impairment among the ASD group ($t = 1.18$, $p = 0.24$, $M = 7.50$ vs. 7.81, $SD = 1.51$ vs. 2.01). Detailed demographic and descriptive statistics for the validation sample are reported in the validation columns of Table 1.

**Frame-level accuracy**. The precision and recall (PR) performance of the deep learning model is illustrated as a blue PR curve in Fig. 2. Each yellow dot in the figure gives the PR for one of the ten expert raters. This PR is obtained by comparing an expert's ratings to the consensus ratings of the other nine experts (effectively treating the nine experts' consensus as ground truth). The mean rater (green diamond) is the average of the PRs for the human raters. The red diamond gives the PR of the model following smoothing (post-processing). In Fig. 2, (a) shows the aggregate PR curve on all 18 validation sessions, and (b), (c), (d) show this curve split by protocol type, gender, and diagnostic category, respectively.

At testing time (including the coder-level reliability and study reproducibility experiments), we used a decision threshold of 0.9 in order to predict the presence or absence eye contact in each frame based on the softmax output of the deep model classifier. In addition, we performed temporal smoothing as a post-processing step in order to reduce noise caused by unwanted events such as face detection failure, motion blur, and eye blinks, we remove outliers and merge short segments through a sliding window scheme using moving average. The classifier decision threshold and the window sizes for outlier removal (5) and merging (6) were chosen via grid search on a held-out training sample, by maximizing detection accuracy while minimizing the event-level eye contact count difference between the estimate and human coding. Note that it achieves slightly higher precision than the mean rater for the same recall. Additionally, we analyzed the impact of face detection on eye contact retrieval accuracy, which we report in the Supplementary Table 1.

Table 2 quantifies the benefits of smoothing and transfer learning. Comparing the F1 scores demonstrates the equivalence of the mean rater and deep model performance. Comparing the third and fourth column, we see that smoothing gives an F1 score increase of 0.01. Similarly, removing transfer learning in the fifth row causes a decrease in F1. We note that the multi-task learning approach from ref. [24], presented in column 6, causes a drop of 0.034 in F1 relative to the transfer learning result in column 3. These results validate the superiority of transfer learning over multi-task learning for this problem.

The average precision can be interpreted as the area under the PR curve, and gives an overall measure of the effectiveness of the classifier without the need to select a particular operating point. In Table 2, we report results for specific operating points. Here we provide the average precision for the classifier without performing smoothing: ESCS 0.948, BOSCC 0.959, and combined 0.956.

**Reliability with human raters**. Inter-rater reliability is measured by Cohen's $\kappa$ for all pairs of human coders and provides evidence for the reliability of the raters. Using the combined dataset, the

| | TD young children Total | TD young children Validation | ASD young children Total | ASD young children Validation | ASD child/adolescent |
|---|---|---|---|---|---|
| **Table 1 Demographics and descriptive statistics.** | | | | | |
| $N$ | 55 | 9 | 66 | 9 | 15 |
| Males | 36 (65%) | 5 (56%) | 55 (83%) | 5 (56%) | 12 (80%) |
| Age (months) | 27.45 (5.84) | 28.78 (5.87) | 44.00 (11.11) | 46.67 (12.25) | 95.56 (32.60) |
| Score | | | | | |
| CSS total | N/A | N/A | 7.97 (2.19) | 7.5 (1.41) | 7.43 (1.74) |
| CSS SA | N/A | N/A | 7.49 (2.30) | 7.5 (1.51) | 7.64 (1.74) |
| CSS RRB | N/A | N/A | 8.19 (1.76) | 7.25 (1.83) | 6.79 (2.83) |
| CBCL total $T$ score | 43.16 (9.02) | 45.43 (10.24) | 58.05 (15.35) | 56.13 (13.28) | 59.00 (8.88) |
| Race | | | | | |
| White/Caucasian (%) | 60 | 89 | 61 | 33 | 60 |
| Black/African | 22 | 11 | 4 | 11 | 0 |
| American (%) | | | | | |
| Asian/Pacific Islander (%) | 0 | 0 | 15 | 33 | 7 |
| More than one race (%) | 13 | 0 | 14 | 11 | 26 |
| Other/unknown (%) | 5 | 0 | 6 | 11 | 7 |
| Hispanic/Latino ethnicity | 4 (7%) | 0 (0%) | 15 (23%) | 3 (33%) | 3 (20%) |
| Maternal education | | | | | |
| Some high school | 1 | 1 | 0 | 0 | 0 |
| High school diploma/GED | 3 | 0 | 3 | 0 | 2 |
| Some college | 8 | 0 | 4 | 1 | 0 |
| College/technical degree | 24 | 4 | 28 | 3 | 5 |
| Graduate school degree | 19 | 4 | 30 | 5 | 8 |
| Unknown | 0 | 0 | 1 | 0 | 0 |

Age, CSS, and CBCL scores expressed as mean (standard deviation). Higher scores indicate more significant impairment/problem behavior. For young children samples, total sample and its validation sub-sample are both reported.
CSS total: calibrated severity score from ADOS; CSS SA: ADOS calibrated severity score for social affect; CSS RRB: ADOS calibrated severity score for restricted and repetitive behaviors; CBCL total $T$ score: score for internalizing and externalizing problem behavior.

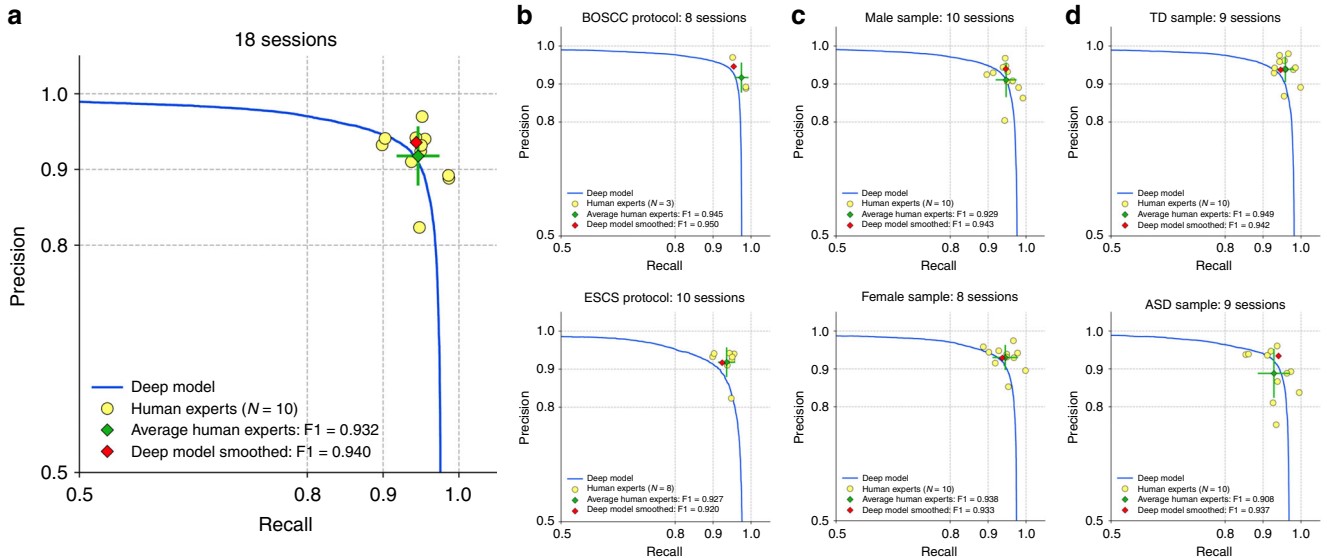

**Fig. 2** Precision and recall (PR) of deep learning model and human raters. The blue line is the PR curve for the model, zoomed into the range 0.5–1.0. Human rater data are presented as mean values ± SD. Improved model PR (red diamond) is obtained by temporally smoothing the model output. The PR for each of the ten expert raters (yellow dots) is obtained by comparing an expert's ratings to the consensus ratings of the other nine experts. **a** PR curve on all 18 validation sessions. The model (red diamond) achieves higher precision than the average of the expert raters (green diamond) for the same recall. The model PR (red diamond) lies within one standard deviation (green error bars) of the mean rater, and both the model and the mean rater have similar F1 scores. Therefore, we conclude that the deep learning model exhibits comparable performance to expert human raters. **b** PR curves computed separately for the BOSCC (top) and the ESCS protocol (bottom). **c** PR curves computed separately for male (top) and female (bottom) samples. **d** PR curves computed separately for TD (top) and ASD (bottom) samples. In all cases, model PR lies within one SD of the mean rater.

| Table 2 Frame-level performance comparisons. | | | | | |
|---|---|---|---|---|---|
| **Metric** | **Mean rater** | **Deep model (smoothed)** | **Deep model (not smoothed)** | **Deep model Without transfer learning (smoothed)** | **Multi-task learning[24] With ResNet (smoothed)** |
| F1 | 0.932 | 0.940 | 0.930 | 0.916 | 0.906 |
| Precision | 0.918 | 0.936 | 0.924 | 0.917 | 0.924 |
| Recall | 0.946 | 0.943 | 0.937 | 0.915 | 0.890 |

Performance without smoothing (fourth column) is reported at the maximum F1 score along the PR curve, with associated PR. In the sixth column, we replaced AlexNet in ref. [24] with ResNet for a fair comparison.

average human–human $\kappa$ is $m_{hh} = 0.888$ (with 0.8 as the standard cut-off for reliability). Comparison of the eye contact detection model with human raters also indicates reliability, with an average human–detector $\kappa$ of $m_{hd} = 0.891$. This demonstrates that adding the model-based detector as an additional "rater" to the pool of human coders preserves reliability, reinforcing the claim that the model is equivalent to a human expert. This hypothesis can be tested statistically using two one-sided tests. For the combined dataset, we are able to reject $H_0: m_{hd} - m_{hh} < -\Delta$ at $p = 0.05$ and accept $H_1$ that the algorithm is as reliable as human annotators, with the equivalence boundary $\Delta$ as low as 0.025, which is the standard deviation of the human $\kappa$'s. We choose this bound as the Smallest Effect Size of Interest (SESOI)[29] as any equivalence range smaller than this is too small to matter.

Figure 3 illustrates the Cohen's $\kappa$ statistics for the human–human and human–detector comparison, and are summarized in Supplementary Table 2.

In the standard two one-sided tests, sample means from the two groups are compared for both sides of inequalities. However, we are only interested in testing if the detector is not less reliable as human. Therefore, the null and alternative

hypotheses tested in this analysis is the following. As shown in Supplementary Table 3, we are able to reject the null hypothesis $H_0$ at $p = 0.05$ for all cases.

$$H_0 : m_{hd} - m_{hh} < -\Delta \tag{1}$$

$$H_1 : -\Delta < m_{hd} - m_{hh} \tag{2}$$

**Reproducibility of prior studies.** Supplementary Fig. 1 provides a visualization of the eye contact frequency and duration rate for each subject, comparing the detection model and human raters. The figure demonstrates good qualitative agreement across subjects, with the model estimates consistently falling within the range defined by the human coders. Note that some subjects are harder to rate, resulting in a greater spread of measures. We replicated the hypotheses tests for significance from the prior studies using the automated coding results. Automated findings were identical to those obtained from human coding. Table 4 summarizes the findings for study[30], with significance for the effect of context on eye contact duration and frequency, but

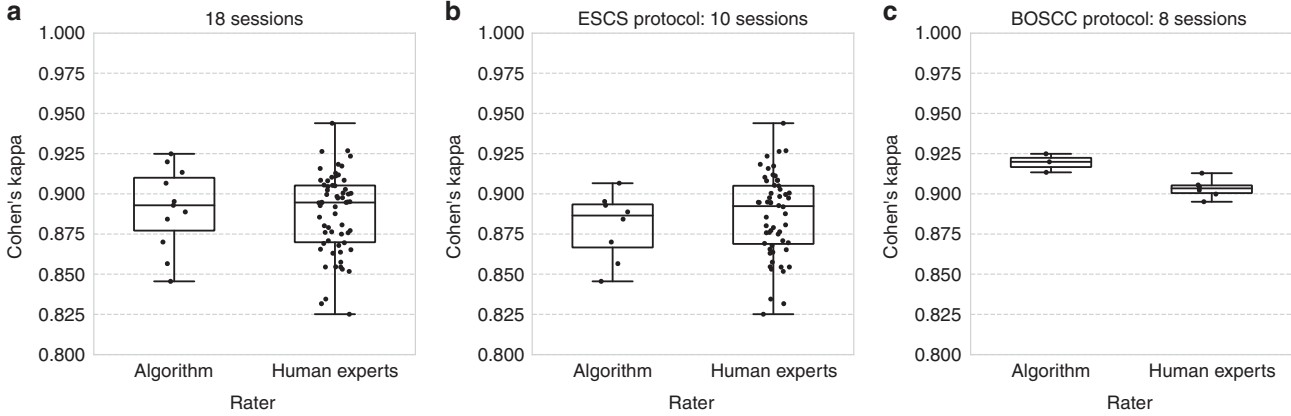

**Fig. 3** Pairwise Cohen's kappa distributions among all human pairs and human–algorithm pairs, represented as box plot. **a** 18 validation sessions, **b** ESCS, **c** BOSCC. Generally, kappa scores above 0.8 are considered an almost perfect agreement. On all sessions annotated by ten human experts, agreements among humans and agreements between each human and algorithm are similar in terms of kappa values.

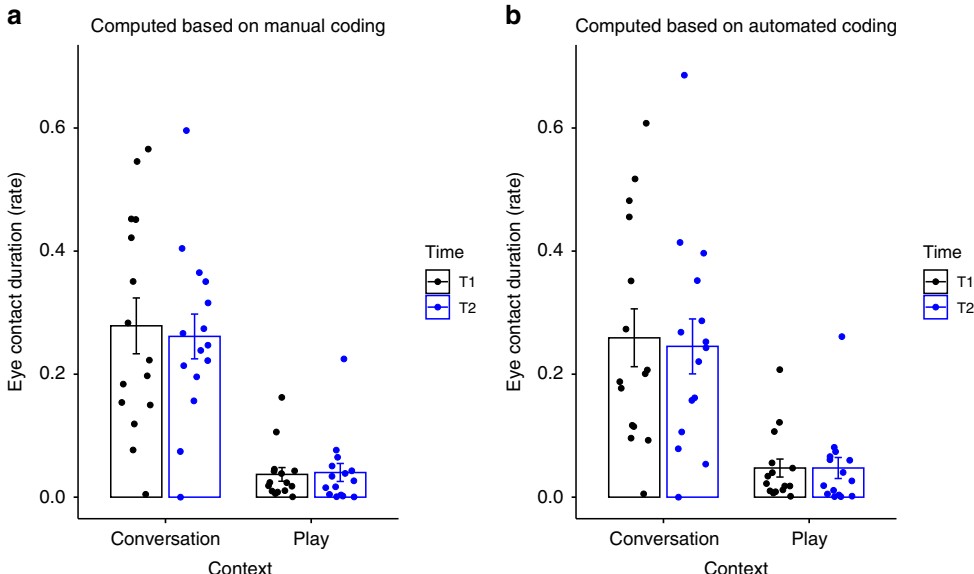

**Fig. 4** Average duration of eye contact during conversation and interactive play in child and adolescent samples ($n = 15$), measured at time 1 and time 2. **a** based on human coding, **b** based on automated coding. Data are presented as mean values ± SEM.

not for the effects of time or time–context interaction. Table 3 and Fig. 4 give the results for study mentioned in ref. [23]. Note that all of the subjects used in this analysis were excluded from the model training set.

**Correlation analysis.** For individuals with ASD, ADOS CSS SA demonstrated a relation with frequency and duration of automatically measured direct gaze during the ESCS ($n = 45$; frequency: $r = -0.41$, $p < 0.01$; duration: $r = -0.36$, $p < 0.05$) and during the BOSCC ($n = 58$; frequency: $r = -0.26$, $p < 0.05$; duration: $r = -0.29$, $p < 0.05$), which was mostly driven by a small number of subjects with low social affect severity scores. Severity of overall social symptoms during the BOSCC (BOSCC SA) demonstrated a strong correlation with both frequency and duration, illustrated in Supplementary Fig. 2 ($n = 25$; frequency: $r = -0.75$, $p < 0.001$; duration: $r = -0.78$, $p < 0.001$).

**Discussion**

This study provides multiple, converging sources of evidence that a deep learning model can detect moments of eye contact in PoV camera video with reliability equivalent to expert human raters. Our model enables the scalable measurement of eye contact during face-to-face interactions. Our findings suggest that it is now feasible to use automated analysis as a substitute for human coding in application domains ranging from autism[23,30] to job interviews[31,32]. We evaluated our method on a diverse dataset containing young children and adolescents from both typical and ASD populations. We believe these are the first findings of equivalence in PR between an automated eye contact detector and human raters.

We now summarize the evidence supporting equivalence between the deep model and expert raters. First, frame-level PR performance for the combined dataset, illustrated in Fig. 2, demonstrates that the smoothed deep model result (red diamond) achieves higher precision than the mean human rater (green diamond) at the same recall. Note also that both the red diamond and the unsmoothed PR curve (in blue) lie within one standard deviation (green error bars) of the mean rater. Second, reliability analysis provides additional evidence to complement the PR analysis results. Treating the deep model as an additional rater results in human–detector reliability of 0.891,

while human–human reliability was 0.888. The distribution of kappas is illustrated in Fig. 3. The hypothesis that the deep model is as reliable as human raters was examined using two one-sided statistical tests and found to hold with an equivalence threshold as low as 0.025. The third source of evidence comes from correlation analysis between eye contact frequency and duration and ASD symptom severity. We would expect increased severity to be negatively correlated with rates of eye contact, and this is born out in Supplementary Fig. 2 which illustrates a strong negative correlation between automatically derived eye contact measures and the BOSCC Social Affect (BOSCC SA) scores. Weaker correlation was found between the derived measures and the ADOS CSS SA scores. The fourth source of evidence comes from a reproducibility study which asks whether the findings from two recently published studies of eye contact in autism continue to hold if automatically derived measures are used in place of expert ratings. Tables 3, 4 and Fig. 4 demonstrate that all findings hold, providing direct evidence for the feasibility of using automatically derived eye contact measures in developmental studies.

An additional contribution of this work is to explore the relative merits of transfer learning and multi-task learning approaches in learning effective models for eye contact. As shown in Table 2, the proposed transfer learning approach that first learns 3D head pose and gaze and then learns eye contact is superior to both the previously proposed multi-task approach[24] that simultaneously learns head pose and eye contact and a baseline method that simply learns eye contact without learning 3D head pose models. Note that if we were determining eye

contact using an external room camera, then the estimation of head pose and gaze angle in 3D would be vitally important to determine where the subject is looking in space. By locating the camera on the subject, we reduce this global 3D estimation task to the much simpler task of assessing gaze relative to the coordinate frame of the camera. Nonetheless, the need to make angular determinations may explain why pretraining on an explicit 3D estimation task leads to improved performance.

We briefly review relevant prior work to place our contribution in context. First, a variety of recent works have demonstrated the feasibility of using deep learning to achieve expert level analysis of biomedical data[33] for detecting and classifying clinical conditions such as diabetic retinopathy[34], skin cancer[35], malignant mammographic lesions[36], bone fracture[37], and atrial fibrillation[38]. In contrast, only a few prior works have explored the automated analysis of social behaviors in clinical contexts such as autism. Early works on automatically analyzing social behaviors[39–42] predated the development of deep learning technology and did not address the issue of expert level performance. Prior work on automatically measuring "response to name" behaviors[43,44] included both ASD and typical samples and assessed the agreement with expert human raters, but did not address naturalistic face-to-face social interactions.

Marinoiu et al.[45] use deep learning models to analyze interactions between children with ASD and a robot therapist, but do not address expert-level performance. Note that none of these prior works addressed the assessment of eye contact. A final line of related work uses machine learning tools to improve the

**Table 3 Original and reproduced statistical tests of ref. [23].**

| | Manual coding p-Value | Manual coding Effect size | Automated coding p-Value | Automated coding Effect size |
|---|---|---|---|---|
| Duration: effect of context | <0.001* | 1.99 | 1.7e−07* | 1.57 |
| Duration: effect of time | >0.9 | | 0.84 | |
| Duration: interaction of time and context | >0.7 | | 0.839 | |
| Frequency: effect of context | <0.001* | 1.48 | 1.2e−07* | 1.59 |
| Frequency: effect of time | >0.9 | | 0.82 | |
| Frequency: interaction of time and context | >0.7 | | 0.85 | |

Percent duration and rate of eye contact during interactive play and conversation across the two time points were compared in 2 (context: play, conversation) by 2 (time: T1, T2) ANOVAs (n = 15). Effect sizes are calculated using Cohen's d.
*Statistically significant at p = 0.05.

**Table 4 Original and reproduced statistical tests of ref. [30].**

| | Manual coding p-Value | Manual coding Effect size | Automated coding p-Value | Automated coding Effect size |
|---|---|---|---|---|
| Cross-group: TD (n = 38) vs. ASD (n = 21), Finding 1 | 0.001* | 0.36 | 0.007* | 0.32 |
| Cross-group: TD (n = 38) vs. ASD (n = 21), Finding 2 | 0.01* | 0.26 | 0.02* | 0.25 |
| Cross-group: TD (n = 38) vs. ASD (n = 21), Finding 3 | 0.06 | | 0.06 | |
| Within-group: TD (n = 38), Finding 4 | <0.001* | 0.92 | 1.01e−7* | 0.84 |
| Within-group: TD (n = 38), Finding 5 | >0.1 | | 0.421 | |
| Within-group: ASD (n = 21), Finding 6 | <0.001* | 0.53 | 0.007* | 0.54 |
| Within-group: ASD (n = 21), Finding 7 | >0.1 | | 0.293 | |

Independent two-group Mann–Whitney U test is used in cross-group analysis and Wilcoxon Signed-Rank test is used for within-group analysis. Effect sizes are calculated using z statistics. Tests are one-sided. Finding 1: TD more EC during toy inactive than ASD. Finding 2: TD more EC in child possession than ASD. Finding 3: TD more EC during toy active than ASD. Finding 4: More EC during toy inactive than active. Finding 5: More EC in examiner possession than in child possession. Finding 6: More EC during toy inactive than active. Finding 7: More EC in examiner possession than in child possession. EC eye contact.
*Statistically significant at p = 0.05.

usability of conventional eye tracking technology by improving robustness to head movements and minimizing the need for calibration[46–48]. While works such as those mentioned in ref. [46] use deep learning to analyze gaze, they focus on the case of gaze to screens or displays. We believe this is the first work to demonstrate human level performance in automatically assessing a social behavior in a naturalistic face-to-face interaction context.

While the focus of this work has been on the assessment of eye contact in interactions between an unencumbered child and an examiner, our technology could also be applied to the analysis of face-to-face interactions between adults in which each subject is wearing video recording glasses. Additional applications in clinical and social psychology[16–20,49] could potentially benefit from this approach. Moreover, these methods can also support the development of social intelligence for robots, enabling them to interact naturally with people using nonverbal social signals. Mutual gaze and joint attention have been found to have a critical role in conversation, narration, collaboration, and manipulation tasks between humans and robots[50–52]. Our analysis shows that our model could cover the common face-to-face interaction distance ranges (Supplementary Fig. 3). We release the trained models and software from this work to facilitate such future work.

## Methods

**Dataset overview**. The dataset used in this study was collected at two institutions between 2015 and 2018. Neurotypical subjects were recruited at the Georgia Tech Child Study Lab in Atlanta, GA (GT), and subjects with ASD were recruited through the Center for Autism and the Developing Brain in White Plains, NY (CADB). All caregivers provided written consent and the Georgia Institute of Technology and Weill Cornell Medicine IRBs approved the study.

**Data collection setup**. Each subject participated in two separate play interactions with a trained examiner. These play interactions, described in more detail below, have been designed and widely used in the psychology research community to elicit nonverbal communication behaviors that are present in typical development, but are often less prevalent in children with autism. Interactions took place at a table with the subject sitting across from the examiner, either on their parent's lap or independently, in order to facilitate data collection. The examiner wore a pair of commercially available camera glasses—Pivothead Kudu (specifications: 1080p video resolution, 30 fps, 3.5-mm-wide angle focal-length, 77° field-of-view)—that provided continuous high resolution capture of the subject's face. The lenses were removed from the glasses to provide an unobstructed view of the examiner's eyes. A stationary camcorder mounted on a tripod was positioned to capture a holistic view of the scene, including the table and all subjects. Our method does not require any camera calibration.

*Play interaction 1: Early Social Communication Scales*. The Early Social Communication Scales (ESCS)[53] is a semi-structured interaction in which the examiner presents a series of social presses involving toys in order to elicit nonverbal communication behaviors (e.g., use of pointing, reaching or eye contact to initiate joint attention or to request a toy). Administration of the ESCS was slightly modified for the study; as eye contact, rather than other social communication behaviors, was the main focus of the current study, ESCS administration was modified in order to elicit more instances of direct gaze. For example, for items that require the child to make a bid to the examiner in order to obtain the toy, examiners would give the toy only when the child made eye contact, not in response to other bids such as pointing or verbally asking. The exception was if the child lost interest in the toy or became agitated prior to making eye contact, the examiner would give the child the toy without requiring any bid. Administration was also modified to remove materials that are meant to be used close to the examiner's face (i.e., hat, comb, and glasses), to prevent ambiguity as to whether the child is looking at the examiner's eyes or at the toy. The ESCS was administered to children 60 months of age and younger.

*Play interaction 2: Brief Observation of Social Communication Change*. The Brief Observation of Social Communication Change (BOSCC)[54] is a naturalistic play-based interaction between the subject and their interaction partner. Subjects in this study completed a modified version of the BOSCC in which they sat at the table rather than on the floor. The BOSCC consisted of two, 4-min segments of free play with toys selected by the child that were followed by two, 2-min segments of snack or conversation, depending on the age of the subject. For the play segments, the subject was presented with a box of toys and asked to select one toy. If the subject did not select a toy, the examiner selected one for themselves and the subject to play with. The subject was only permitted to have one toy on the table at a time, but the subject could select a different toy at any time. During snack segments, the examiner presented two clear containers with different snacks, and the subject selected which snack they would like. Children were given small portions of snack in order to create opportunities for additional requesting. The BOSCC is designed to measure change over time, and its inclusion in our dataset serves to increase the diversity of the gaze behavior contexts.

**Participants**. A total of 66 children (55 male) who were suspected of having ASD were recruited at CADB and 58 typically developing (TD) children (36 male) were recruited at Georgia Tech to participate in a study designed to validate the feasibility of using PoV glasses to capture gaze behavior in young children (18–60 months, $M = 36.48$ months). Subjects who were suspected of having ASD were evaluated with the Autism Diagnostic Observation Schedule (ADOS-2)[55]. TD subjects were screened for developmental delays with the CSBS-DP[56] or M-CHAT[57]. Three TD subjects were excluded from all analyses due to technical issues. All subjects completed both the ESCS and the BOSCC in randomized order, regardless of their diagnostic status. A subset of subjects ($n = 14$) completed a follow-up session within a year of their first visit, and another subset ($n = 27$) completed an additional BOSCC with their parent as the examiner. Eight subjects completed a parent-led BOSCC at their follow-up session. (Additional follow-up or parent-led BOSCCs were administered as part of a pilot data collection for another project and we chose to include these data in our analysis to have a wider variety of contexts from which we have sampled children's gaze behavior.) In total, the number of sessions is 167. All subjects in the young children sample, regardless of diagnostic status, were utilized in training and validating the eye contact model. Thus, the Young Children dataset consists of 66 ASD and 55 TD children (see Table 1, columns 2 and 4, for detailed demographic information).

Three subjects scored in the low range of concern for ASD and one subject did not complete the ADOS-2 and these four subjects were not included in analyses comparing ASD to TD groups. Three TD subjects were excluded from analyses comparing across groups because of concern for developmental delay.

The BOSCC social affect scores are only available for the 25 minimally verbal subjects within ASD group as they are considered appropriate for scoring[54].

**Video coding of eye contact**. Mangold International's INTERACT video annotation software (https://www.mangold-international.com) was used by coders to flag frame-level onsets and offsets of eye contact during ESCS and BOSCC protocols. The ESCS sessions from 10 subjects (5 TD, 5 ASD) were annotated by 8 independent raters to establish reliability (mean $\kappa = 0.886$). In addition, 3 raters (1 rater in common with the first group) annotated the BOSCC videos from 8 subjects (4 TD, 4 ASD), in order to test reliability on a different play protocol (mean $\kappa = 0.903$). The remaining ESCS and BOSCC sessions were annotated by single raters. The video segments from the 10 ESCS and 8 BOSCC sessions which were annotated by multiple raters constitute the validation set, which was used to test generalization performance of the gaze detection model after training. The remaining single-rater sessions from 103 subjects comprise the training set. In our dataset, eye contact was annotated for both BOSCC and ESCS assessments for 45 subjects. For the remaining 76 subjects, eye contact annotations were only completed for one of the assessments ($n = 47$ for the ESCS; $n = 29$ for the BOSCC) due to the time-consuming nature of the manual coding process. The validation set has no overlap with the training set and is representative of the total sample in terms of age, race, diagnosis, gender, and autism severity. Note that we have shown in prior work that human coding of eye contact from PoV video can be achieved with greater reliability than from a standard video recorder[22], thus validating our data annotation approach. This prior work also demonstrated that wearing the glasses did not impact the frequency of eye contact bids in a sample of 2-to-4-year-olds.

**Data preparation**. PoV video frames were decoded at 30 frames per second (capture rate) and saved to disk. In each frame, the subject's face was detected and recognized following the procedure from ref. [24]. In the training set, each frame is labeled by a single rater, with 1 for eye contact and 0 otherwise, which was abstracted from the onset–offset coding. In the validation set, each frame has annotations from multiple raters and the majority vote is used as the ground truth label. The datasets consist of 4,339,879 frames (281,152 with eye contact) for training and 353,924 frames (25,112 with eye contact) for validation.

**Training algorithm**. We used a deep convolutional neural network (CNN) with a ResNet-50 backbone architecture[58] as our classifier model (see Fig. 5). ResNet, short for residual network, is a popular deep neural network type, and the 50 in ResNet-50 refers to the number of layers it has. The inputs consist of cropped face regions, resized to 224 × 224 pixels. We used a two-stage training process to support task transfer learning. In the first stage, training on three public datasets enables the model to learn the relationship between head pose and eye gaze direction as follows: the model is trained to regress the 3D gaze direction based on MPIIFaceGaze[47] and EYEDIAP[59] datasets and 3D head pose with the SynHead[60] dataset, using the poss regression method of ref. [61]. Convergence is defined as reaching <6° mean absolute error on gaze angle and head pose. The model is then

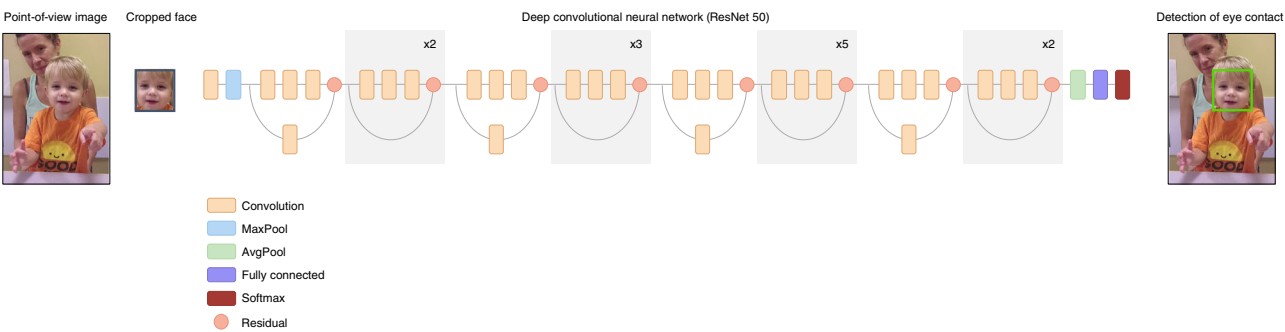

**Fig. 5** Deep neural network layout. Given a frame extracted from a point-of-view camera, the subject's face is automatically detected and cropped as an input to the deep neural networks. A deep neural network is used to compute the features from facial image via a series of convolutions. At the end of the network, features are combined through average pooling and fully connected layers, and the softmax operation produces the final eye contact score. Using this score, algorithm can decide if the input face is an eye contact. The authors have obtained consent to publish the sample picture from the study participants.

fine-tuned using our training dataset, in order to learn the condition of eye contact and capture the details of children's facial appearance.

Parameters are fine-tuned across the last two blocks of the ResNet layers, using cross entropy loss with a re-weighting factor of 0.1 which is multiplied by the loss of the over-represented class in order to compensate for the class imbalance (eye contact presence vs. absence ratio) in our dataset. Backpropagation with a learning rate of 0.005 under Adam optimization is used to update network weights for every mini-batch of 256 samples until it has seen three epochs of training data with augmentation. Data augmentation during training consists of a combination of horizontal flip, color jitter (brightness, contrast, saturation ≤±20%), blur (Gaussian kernel size ≤0.6), and face bounding box re-scale (≤+20%), each of which takes effect at a probability of 0.5. Augmentation parameters are uniformly sampled within the given range. We use PyTorch deep learning framework to train and evaluate our network.

Note that our prior work[24] used an architecture that incorporated multi-task learning, a machine learning approach in which multiple learning tasks are solved at the same time, by forcing the network to predict the head pose and the eye contact label during training. A finding from this work is that the transfer learning approach is more effective.

**Evaluation overview**. We performed three experiments to evaluate the performance of our approach. The first experiment evaluates the per frame prediction accuracy of our method. The second experiment evaluates the inter-rater reliability of the deep learning model with respect to a set of human raters. The third experiment tests whether the direct application of our eye contact detector can replicate findings about eye contact behavior in two previously-reported studies which used manual coding. These experiments provide three sources of converging evidence for our primary hypothesis: That automatic detection of eye contact via a deep learning classifier yields performance which is equivalent to the accuracy of human coders, making automatic video coding viable for research studies in social communication. Each analysis uses a different subset of subjects as it is considered appropriate based on the availability of diagnostic category and assessment scores, as shown in Supplementary Fig. 4.

**Evaluation for frame-level accuracy**. We compute the precision–recall (PR) curve on the validation dataset as a function of the classification threshold, using the majority vote of the human coders as the ground truth. We summarize the PR curve using maximum F1 score and average precision which are defined as follows:

$$\text{Precision} = \frac{\text{Truepositive}}{\text{Truepositive} + \text{falsepositive}}, \quad (3)$$

$$\text{Recall} = \frac{\text{Truepositive}}{\text{Truepositive} + \text{falsenegative}}, \quad (4)$$

where 'true positive' is the number of correctly predicted eye contact, 'false positive' is the number of incorrectly predicted non eye contact, 'false negative' is the number of incorrectly predicted eye contact.

As the neural network outputs a score $S$ per frame (taken from the softmax layer), a PR curve is generated by choosing a fixed threshold score $t$ such that the prediction $\hat{y}$ for each image is defined as $\hat{y} = S \geq t$, then sweeping $t$ in the interval 0–1. Due to the imbalance between the two classes (eye contact vs. other) in our dataset, PR curve is better suited for evaluation than receiver operating characteristic, as PR does not take into account abundant true negatives.

Maximum F1 score = $\frac{2 \times P_t \times R_t}{P_t + R_t}$ and average precision (AP) = $\sum_t (R_t - R_{t-1}) \times P_t$ are also reported as summary statistics for PR curve ($P_t$ and $R_t$ are the precision and recall, respectively, at threshold $t$).

**Evaluation for reliability with human raters**. For this evaluation, we threshold the model's output and perform post-processing to predict an eye contact label for each frame of an input video sequence. We treat the learned model as an additional (automated) video coder, and compute the agreement between the algorithm and the human raters on the validation dataset. We use Cohen's kappa score[62] to measure the inter-rater reliability between the model and each of the raters, and between all pairs of raters, where an average kappa score greater than 0.8 is usually considered to be sufficient to establish reliability for a group of raters.

In addition, we use two one-sided tests (TOST)[63] to statistically test the hypothesis of equivalence between the human annotator group and the automatic coding algorithm. In TOST, the null hypothesis is a sample mean difference greater than Δ, and the alternative hypothesis is equivalence between the classes:

$$H_0 : m_{hd} - m_{hh} < -\Delta \quad \text{or} \quad m_{hd} - m_{hh} > \Delta, \quad (5)$$

$$H_1 : -\Delta < m_{hd} - m_{hh} < \Delta, \quad (6)$$

where $m_{hd}$ is the mean of kappa scores of all human–detector pairs, $m_{hh}$ is the mean of Cohen's kappa scores between all human pairs, and Δ is the equivalence boundary.

**Replication of prior studies**. Our final evaluation assesses the impact of replacing human coding with computer coding of eye contact in prior observational studies. We repeat the data analysis for the two studies described in refs. [23,30], using eye contact statistics obtained from applying our automated method to the original video files, as an alternative to the manual coding originally performed by the authors. For both studies, we compute eye contact frequency (eye contact event counts per minute) and duration rate (eye contact duration divided by the administration time) using the algorithm's output, and repeat the statistical tests as in the original analyses.

The study in ref. [30] examined differences in eye contact rates between ASD and TD young children during the toy–spectacle tasks of the ESCS in consideration of temporal-contextual factors such as activation and possession of the toy. Samples used in training the eye contact model overlap partially with the samples used in ref. [30]. In order to eliminate any potential bias, we removed the overlapping subjects ($n = 47$) from the training set and retrained the eye contact model on 56 subjects (instead of 103) for use in this experiment.

The study in ref. [23] investigated increased eye contact in individuals with ASD during conversation as compared to play in the BOSCC protocol in an additional group of 15 older children and adolescents with ASD (three females, 5–13 years, $M = 8$ years). Table 1, last column, provides the detailed demographic information. We reproduce the analysis of 'Sample 2' from ref. [23] only, as 'Sample 1' was not available.

**Correlation analysis**. We performed an additional experiment to examine the correlation between automatically measured eye contact statistics (frequency and duration rate) and symptom severity, using the sample of ASD subjects aged up to 60 months. Given the importance of eye contact for overall social communication skills in young children[5–7], it was expected that frequency and duration of eye contact would demonstrate a negative correlation with social impairment among the autism subjects. Note that some parts of the videos in our data were not manually annotated for eye contact, and using automatically measured eye contact statistics is a scalable and favorable way of testing this hypothesis.

Symptom severity was assessed using two social affect scores derived from the ADOS and the BOSCC. First, the ADOS Calibrated Severity Scores for Social Affect (ADOS CSS SA) were computed from children's ADOS[64] assessments. In addition, the BOSCC Social Affect (BOSCC SA) scores were coded for 25 subjects who were minimally verbal[54]. These were calculated by summing scores on items 1–9 for each of the two BOSCC segments and averaging the totals. For both scores, higher

numbers indicate greater severity of social impairment. BOSCC SA scores for the remaining 41 subjects in the ASD group were not assigned as the subjects had more verbal language than the minimally verbal level for which the BOSCC is validated, as indicated by the subject having completed an ADOS-2 module 2 or 3, or speaking in flexible phrases or sentences during the BOSCC.

Pearson correlation coefficients were computed between ADOS CSS SA and the automated eye contact measures (frequency and duration rate) from the ESCS segment and the BOSCC segment (separately). In addition, correlations were computed between the BOSCC SA and the automated eye contact measures (frequency and duration rate) from the BOSCC segment.

All subjects considered for correlation analysis are a subset of ASD young children samples. Since they were part of the training set, we use the reduced model that is used for reproducibility analysis for[30] that is trained with 56 subjects, in order to minimize bias. We avoid dropping additional subjects further from this as it would cause shrinking the training set too much and degrade its performance. As a result, subjects included in correlation analysis are partly represented in the sample that the model was trained with. Namely, there are $n = 58$ (39 trained) subjects for ADOS CSS SA correlation during the BOSCC, $n = 45$ (25 trained) subjects for ADOS CSS SA correlation during the ESCS, and $n = 25$ (16 trained) for BOSCC SA correlation analysis.

**Reporting summary**. Further information on research design is available in the Nature Research Reporting Summary linked to this article.

## Data availability
Three datasets that were used in the first stage of training are publicly available; MPIIFaceGaze: https://www.mpi-inf.mpg.de/departments/computer-vision-and-machine-learning/research/gaze-based-human-computer-interaction/its-written-all-over-your-face-full-face-appearance-based-gaze-estimation. EYEDIAP: https://www.idiap.ch/dataset/eyediap. SynHead: https://research.nvidia.com/publication/dynamic-facial-analysis-bayesian-filtering-recurrent-neural-networks. The IRB protocol for this project prohibits the release of the eye contact dataset itself.

## Code availability
The deep learning model trained for this manuscript and the associated test code can be found at https://github.com/rehg-lab/eye-contact-cnn.

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

## Acknowledgements

This study was funded in part by Simons Foundation awards 336363 and 383667, and NIH R01 MH114999. We thank Benjamin Silver, Erin McDonald, Ellise Sims, and Sarah Nay for their contributions to data annotation.

## Author contributions

E.C., C.L., A.R., R.M.J., and J.M.R. participated in the design of the study; E.C.W., A.S., E.S., C.M., E.L.A., M.R.S., and A.R. collected and coded the data; E.C. and E.C.W. performed data analysis; E.C., E.C.W., A.R., R.M.J., and J.M.R. wrote the paper.

## Competing interests

The authors declare no competing interests.
