## [Peer Review File · Nature Communications]

Reviewers' Comments:

Reviewer #1:

Remarks to the Author:

This paper shows 1) a method of eye contact detection using deep neural net and 2) the performance of the eye contact detection is equal or more reliable than the human coders. Regarding 1), the paper proposes a method using ResNet-based algorithm trained using two different datasets, namely, open dataset such as MPIIGaze and original dataset targeting children with ASD. Regarding the point 2), the method shows the proposed DNN-based algorithm may become a replacement of one of the human coders.

Overall, I think this paper is beneficial for the researchers in both computer science and psychology since it shows the new application scenario of image-based analysis for the people in CS, and new methodology of behavioral analysis for the people in psychology and medical field since they pay a lot of efforts and costs for behavioral coding. The result is convincing showing the power of DNN.

However, I have two concerns regarding the algorithm.

1. The method uses RESNET that inputs whole facial image which have limited image resolution. On the other hand, several other methods use only eye region for gaze or eye-contact estimation. Do authors think the current resolution is enough for the task? Discuss the current network architecture is enough for the task.

The second concern is that the proposed approach does not use temporal inference though it is quite important for the human communication analysis. Discuss why single-frame inference is enough for the task.

2. Experimental setup of first person video capturing.

The method do not describe the video capturing setups of the first person camera and specification of the camera such as image resolution and viewing angles, which is critical for reproductivity. Discuss the method does not limited for the particular experimental setups – such as cameras or experimental setups.

In addition, I am concerning the relation between the camera wearer's viewing direction (straight gaze direction) and the image coordinate in the first person video. Ideally, the straight gaze direction should be located at the center of the image coordinate. Did you perform any calibration step for the camera setup? If so, describe the condition of the video capturing.

3. Image resolution and distance to the subjects.

Since the method employ fixed resolution for the input image, the accuracy of the eye contact detection depends on the original size of face which is related to the distance to the subjects. Discuss and show the relation between the accuracy v.s distance to the children (size of the children's faces). Ideally, it should be shown the accuracy comparison of the human annotator and DNN in the sense of the distance to the face and the recognition accuracy.

Reviewer #2:

Remarks to the Author:

The paper employs a deep learning algorithm for estimating a subject's visual focus of attention (VFoA), specifically designed to aid diagnosis of Autism Spectrum Disorder (ASD). Based on the authors' claims, the network's performance is on par with 10 trained human coders, indicating that the system reliability is comparable to humans. While the paper presents a viable solution to a topical problem, there are multiple issues with the paper which need addressing, and requiring appropriate modifications. I am outlining my concerns with the manuscript below:

(1) Firstly, the claims made by the authors are overreaching. To say that their algorithm is "the first deep network model to detect eye contact in ego-centric video" seems misleading and incorrect. Firstly, the deep network model employed in the work is the well-known ResNet50, and not a novel contribution.

Secondly, based on the information presented in Figures 1 and 2, as well as the descriptions of the two play interactions, it appears that the interactions were more or less controlled, and the problem involved monitoring the subject of interest (children) for eye contact. This is not necessarily more challenging than other other works targeting a clinical/social application-- some these works (such as the ones listed below) should be acknowledged and contrasted with by the authors apart from their own prior works:

a) Viral Parekh, Pin Sym Foong, Shengdong Zhao, and Ramanathan Subramanian. 2018. AVEID: Automatic Video System for Measuring Engagement In Dementia. In 23rd International Conference on Intelligent User Interfaces (IUI '18). Association for Computing Machinery, New York, NY, USA, 409–413.

b) Samira Sheikhi, Jean-Marc Odobez, Combining dynamic head pose-gaze mapping with the robot conversational state for attention recognition in human-robot interactions, Pattern Recognition Letters, Volume 66, Issue C, 2015.

(2) Given that the intended audience of this paper is multi-disciplinary (AI+psychology), it should be easily understandable to the non-specialist! In this regard, all terms used in the paper should be clearly introduced. However, this is not the case. E.g., the term 'Res Net 50' is introduced in the caption of Figure 2 without any reference or description. Similarly, the term 'Multi-task learning' is not introduced before being used on Page 15.

(3) Why is it significant/necessary to employ two play interaction types in the study? Do these represent different settings with respect to gaze estimation or they are two distinct settings to study ASD? This should be clarified at the outset.

(4) The facial cues employed to detect eye contact are unclear. The MPIIGaze and EYEDIAP datasets include high-resolution face images, from which both head pose and eye-based gaze estimation can be performed. This apparently seems to be the case based on the information presented on Page 7. However, the algorithms employed for face and eye detection are unspecified. The GazeFollow network used in reference (a) requires the user to input a rectangular bounding box on the subject's face to circumvent difficulties with automated face detection. What is the impact of face and eye detection accuracy on the proposed system performance (I suspect it would make the system less reliable than human performance)? This discussion needs to be presented.

(5) One of my biggest issues with the paper concerns the training algorithm. Page 7 mentions that a regression model is trained based on MPIIGaze and EYEDIAP, while the fine-tuning involves minimizing the cross entropy loss, which is defined for classification models. The performance metrics employed (Precision, Recall and F1-score) are also classification metrics. From the information presented, it appears that the annotations acquired also related to whether the subject made eye contact with the examiner or not (classification).

So, this disconnect between classification and regression-based model training is highly confusing, and raises doubts over the credibility of the presented results.

(6) Why are studies [40,23] critical to this work, and why do they need to be replicated (Page 8,9)? When they are not baselines used to evaluate eye contact detection against this work, I would recommend detailing synthesis of train and test sets, as well as the experiment independent of the reference to these works.

(7) What are the methodologies employed for outlier removal and merging on Page 10?

(8) I would recommend consistent use of terminology throughout the paper-- e.g., social vs interactive partner (Page 2). It appears from Figure 2 that interactive partner is the appropriate term as the parent is also the child's social partner but not a subject of interest.

(9) Also, inconsistent placement of figures in the paper. E.g., Fig.1 & 2 are inline, while Fig. 3--6 are placed at the end.

Reviewer #3:

Remarks to the Author:

Review : Detection of eye contact with deep neural networks is as accurate as human experts. In this study, the authors explore the possibility of achieving accurate detection of eye contact recorded during periods of social interaction with children of varying ages, both typical and with autistic spectrum disorders, through an artificial neural network. The interactions between the observers and children took place during the use of tests to assess socio-relational (ESCS, BOSCC) skills in subjects screened for neurodevelopmental disorders. Interesting they used videos acquired from an egocentric point view through a cheap camera mounted on eye-glasses, acquired also from past studies of the same authors. The results showed that deep learning model can produce automated coded comparable with those produced by human coders, showing a high level of concordance (Kappa scores) and similar overall precision. Authors finally suggest that this could be a novel method to analyze gaze behavior in naturalistic social setting, and represent a cheap method to help research and clinicians during assessment of neurodevelopmental disorders.

Evaluation

This study uses a robust design with very strong data, communicating the main points very well. This is an important study because there are not many studies available to test the possibility of using an automatic eye-contact recognition method comparable to that of human raters, which can support clinical diagnosis. I think current study is publishable provided some minor clarification

Below, I provide some suggestions:

#1 in the Participants section the text would benefit from the presence of the demographic table, now included in the supplementary material, with the related statistical comparisons. It would also be useful to include the sub-sample used for the validation set within this table, with related comparisons with the total group, in order to support the text in the results section (pg 10).

#2 Data Collection setup : It is not clear from the text which subjects have been assessed with the ESCS scale and which ones with the BOSCC and which criteria have been chosen to carry out the assessment. Instead, it seems to me that for the ADOS scale all subjects with a suspicious autism spectrum disorder have carried out the evaluation. These information should be reported in the text or in table. Could the authors provide it?

3# Results :

Page 10 see point 1. Please provide Mean and STD of the statistical comparison cited in the text. Effect size are missing in the analysis, please the author provide it.

Did you look at gender differences in the data?

Correlational Analysis. Would be useful show if also the frequency and duration of eye contact, coded by human raters are correlated with the clinical assessment. In general from a clinical point of view would be interesting see if this automated coding is able to show differences in the frequency and duration of eye contact able to show a difference between typically children compared with the ASD group, in order to find potential biomarkers.

Reviewer 1

Reviewer 1-1a The method uses RESNET that inputs the whole facial image which has limited image resolution. On the other hand, several other methods use only eye region for gaze or eye-contact estimation. Do authors think the current resolution is enough for the task? Discuss the current network architecture is enough for the task.

Reviewer 1-3 Since the method employs fixed resolution for the input image, the accuracy of the eye contact detection depends on the original size of face which is related to the distance to the subjects. Discuss and show the relation between the accuracy v.s distance to the children (size of the children's faces). Ideally, it should be shown the accuracy comparison of the human annotator and DNN in the sense of the distance to the face and the recognition accuracy.

Author Response: Thank you for this question. We have conducted additional analysis to address this issue. The key observation is that our work is motivated by face-to-face social interactions which take place over a relatively limited range of interpersonal distances. We studied the effect of varying this distance on the performance of our model. Increasing the interpersonal distance has the consequence of reducing the size of the target face in the input image. We conducted an additional simulation analysis by down-sampling the face images that are input to our model, to capture the presentation of faces over a range of distances. We found that the accuracy of our model is unaffected by interpersonal distances of up to 95cm (Eye contact F1 score: model = 0.940, average human = 0.932). Moreover, performance is minimally effected by distances of up to 2.3 meters (F1 score: model = 0.931). This distance range is more than adequate to cover scenarios of face-to-face interaction. For example, this range covers the zones of intimate and personal space defined in the study of proxemics (Hall, 1963) in which person-to-person social interactions occur.

[1] Hall, E. T. (1963). A system for the notation of proxemic behavior. *American anthropologist*, 65(5), 1003-1026.

Reviewer 1-1b The proposed approach does not use temporal inference though it is quite important for the human communication analysis. Discuss why single-frame inference is enough for the task.

Author Response: We agree that temporal cues are an important part of human communication. Although we did not design an explicit temporal model, our proposed method does incorporate temporal smoothing based on a sliding window (described on page 12), which has the effect of incorporating temporal cues to a limited extent. We report that eye contact computed by this

method consistently attains human-level accuracy. We also note that when human annotators are performing gaze coding in video, they routinely use temporal cues, as gaze coding tools support scrubbing through the video and comparing frames over time. Therefore, the gold standard against which we compare performance does involve significant temporal inference.

We acknowledge that eye contact is just one gaze-based communicative signal and more sophisticated temporal inference models may be required for detecting more time-varying gaze cues, such as a gaze shift and joint attention behaviors.

Reviewer 1-2 The method do not describe the video capturing setups of the first person camera and specification of the camera such as image resolution and viewing angles, which is critical for reproducibility. Discuss the method does not limited for the particular experimental setups – such as cameras or experimental setups. In addition, I am concerning the relation between the camera wearer’s viewing direction (straight gaze direction) and the image coordinate in the first person video. Ideally, the straight gaze direction should be located at the center of the image coordinate. Did you perform any calibration step for the camera setup? If so, describe the condition of the video capturing.

Author Response: Thank you for this suggestion. This information has now been included on page 4. *Camera specifications; 1080p video, 30 fps, 3.5mm-wide angle focal-length, 77° field-of-view.* Our method does not require any camera calibration, which is now stated in the manuscript as well.

Regarding the issue of camera alignment with viewing direction: The requirements for camera alignment are quite minimal for our approach, in comparison for example to the situation with wearable eye trackers (that require alignment and accurate calibration), due to the fact that in our case the camera is used to assess the social partner’s gaze, while eye trackers are used to assess the glasses-wearer’s gaze. Small shifts in the position of the glasses on the face, which can invalidate calibration in the case of wearable eye trackers, have negligible affect on the accuracy of our eye contact detection method because of their minimal affect on the determination of where the social partner is looking.

Reviewer 2

Reviewer 2-1 Firstly, the claims made by the authors are overreaching. To say that their algorithm is “the first deep network model to detect eye contact in ego-centric video” seems misleading and incorrect. Firstly, the deep network model employed in the work is the well-known ResNet50, and not a novel contribution.

Secondly, based on the information presented in Figures 1 and 2, as well as the descriptions

of the two play interactions, it appears that the interactions were more or less controlled, and the problem involved monitoring the subject of interest (children) for eye contact. This is not necessarily more challenging than other works targeting a clinical/social application– some these works (such as the ones listed below) should be acknowledged and contrasted with by the authors apart from their own prior works.

Author Response: Thank you for this comment, we have updated the writing to address it. Our key claim is the *first demonstration of human-level performance* of a deep model in detecting eye contact in naturalistic face-to-face interactions. To avoid any appearance of claims for architectural novelty, we have amended the claim in the abstract to “*In this work, we developed a deep neural network model to automatically detect eye contact in egocentric video that is the first to achieve accuracy equivalent to that of human experts.*”.

We also appreciate the suggestions for citing additional related works. They are now incorporated in the Discussion and included in our references.

Reviewer 2-2 Given that the intended audience of this paper is multi-disciplinary (AI+psychology), it should be easily understandable to the non-specialist! In this regard, all terms used in the paper should be clearly introduced. However, this is not the case. E.g., the term ‘Res Net 50’ is introduced in the caption of Figure 2 without any reference or description. Similarly, the term ‘Multi-task learning’ is not introduced before being used on Page 15.

Author Response: Thank you for this suggestion. In the revised manuscript, we have ensured that all technical terms are clearly described the first time they are referenced.

Reviewer 2-3 Why is it significant/necessary to employ two play interaction types in the study? Do these represent different settings with respect to gaze estimation or they are two distinct settings to study ASD? This should be clarified at the outset.

Author Response: We utilized two toy-based play interactions in order to capture social gaze behavior in ASD in two different, commonly-used assessment contexts: a free play context (BOSCC) and a more structured context (ESCS). We updated the text to clarify this point.

Reviewer 2-4 The facial cues employed to detect eye contact are unclear. The MPIIGaze and EYEDIAP datasets include high-resolution face images, from which both head pose and eye-based gaze estimation can be performed. This apparently seems to be the case based on the information presented on Page 7. However, the algorithms employed for face and eye detection are unspecified. The Gazefollow network used in reference (a) requires the user to input a rectangular bounding box on the subject’s face to circumvent difficulties with automated face detection. What is the impact of face and eye detection accuracy on the proposed system performance (I suspect it would make the system less reliable than human

performance)? **This discussion needs to be presented.**

Author Response: We wish to clarify that we do not perform *eye detection*, since the model works with the whole face image as its only input. The face detection process is automatic and done using a standard a Faster-RCNN-based detector, following approach of (Chong, et. al. 2017) as described in the Data preparation section on page 7.

Thank you for the valuable suggestion of quantifying the impact of face detection failure on the accuracy of our approach. To that end, we analyzed the number of frames in which the face detector failed, to quantify the percentage of eye contact detection error which resulted from the failure of the face detector. This data is now included in Supplementary Table 7, also included below. As Table 1 indicates, only 2.34% of eye contact test cases were missed due to a face detection failure.

Table 1: Face detection accuracy analysis. True face in frame counted based on eye contact annotations (1st row), number of eye contact frames with face detection failure (2nd row), % of frames with face detection failure (3rd row).

	train	test	train+test
# of face frames with eye contact	281,152	25,112	306,264
# of eye contact frames with failed face detection	12,420	587	13,007
% of eye contact frames with failed face detection	4.42 %	2.34 %	4.25 %

Reviewer 2-5 One of my biggest issues with the paper concerns the training algorithm. Page 7 mentions that a regression model is trained based on MPIIGaze and EYEDIAP, while the fine-tuning involves minimizing the cross entropy loss, which is defined for classification models. The performance metrics employed (Precision, Recall and F1-score) are also classification metrics. From the information presented, it appears that the annotations acquired also related to whether the subject made eye contact with the examiner or not (classification). So, this disconnect between classification and regression-based model training is highly confusing, and raises doubts over the credibility of the presented results.

Author Response: Thank you for bringing to our attention that the citation for the actual regression model we used during pre-training was missing from the manuscript. The citation has now been added. Our pose regression method leverages the following work:

[2] Ruiz, N., Chong, E., & Rehg, J. M. (2018). Fine-grained head pose estimation without keypoints. In Proceedings of the IEEE conference on computer vision and pattern recognition workshops (pp. 2074-2083).

Figure A: Illustration of the loss function used during pose regression, resulting in a learned feature representation that we use for transfer learning in our approach.

Figure A, taken from this reference and added here for clarity, illustrates the loss function we used. The loss function for pose regression represents a weighted sum of the cross entropy loss over a number of discretized pose bins and the MSE loss. In our experiment, we used balanced weights (cross entropy:MSE = 1:1), meaning half of the gradients were attributed to the classification loss in pre-training.

The need for different loss functions is a consequence of the fact that the label space is different between datasets used in pre-training vs. fine-tuning, e.g., angular value for pose regression vs. specific pose class. Such a setup in which different losses are used to enable transfer learning in a heterogeneous label space was shown to be beneficial prior works (Zhang, et. al. 2019). In the case of CNN-based architectures like ours, it has been shown that layers initially trained for one task can be valuable even for considerably different tasks across a wide range of visual recognition problems (Donahue, et. al. 2014, Sharif Razavian, et. al. 2014). The use of different loss functions in pre-training does not pose a problem in this context. For example, (Erhan, et. al. 2010) demonstrated that unsupervised learning with auto-encoders (regression task with MSE loss) can be used effectively as a pre-trained model to improve digit recognition (classification task with softmax loss).

[3] Zhang, J., Li, W., Ogunbona, P., & Xu, D. (2019). Recent advances in transfer learning for cross-dataset visual recognition: A problem-oriented perspective. *ACM Computing Surveys (CSUR)*, 52(1), 1-38.

[4] Donahue, J., Jia, Y., Vinyals, O., Hoffman, J., Zhang, N., Tzeng, E., & Darrell, T. (2014, January). Decaf: A deep convolutional activation feature for generic visual

recognition. In International conference on machine learning (pp. 647-655).

[5] Sharif Razavian, A., Azizpour, H., Sullivan, J., & Carlsson, S. (2014). CNN features off-the-shelf: an astounding baseline for recognition. In Proceedings of the IEEE conference on computer vision and pattern recognition workshops (pp. 806-813).

[6] Erhan, D., Bengio, Y., Courville, A., Manzagol, P. A., Vincent, P., & Bengio, S. (2010). Why does unsupervised pre-training help deep learning?. *Journal of Machine Learning Research*, 11(Feb), 625-660.

Reviewer 2-6 Why are studies [40,23] critical to this work, and why do they need to be replicated (Page 8,9)? When they are not baselines used to evaluate eye contact detection against this work, I would recommend detailing synthesis of train and test sets, as well as the experiment independent of the reference to these works.

Author Response: These additional replication studies [40,23] (which are independent of the baselines for quantifying the accuracy of eye contact) were added to directly test one of the potential scientific benefits of our work: Namely the ability to test hypotheses about behavior on collected data using the automated detector, *without performing manual coding*. In this situation, the primary threat to validity is that a statistical test for validating a scientific hypothesis would yield a different outcome in the case of human coding vs. automated detection. We chose the studies in question because they are focused on gaze behavior in ASD and we have access to the original captured videos. In the published works, hypothesis testing was based on manually-coded results. In our experiments, we examined how the significance level for the hypothesis test would be affected by the use automated detection, and showed that the use of automated measures led to the same level of significance for the scientific hypothesis.

Reviewer 2-7 What are the methodologies employed for outlier removal and merging on Page 10?

Author Response: We used a moving average scheme. This detail is now included in the text on page 12.

Reviewer 2-8 I would recommend consistent use of terminology throughout the paper– e.g., social vs interactive partner (Page 2). It appears from Figure 2 that interactive partner is the appropriate term as the parent is also the child’s social partner but not a subject of interest.

Author Response: Thank you for this suggestion. We now use the term interactive partner throughout the manuscript.

Reviewer 2-9 Also, inconsistent placement of figures in the paper. E.g., Fig.1 & 2 are inline, while Fig. 3–6 are placed at the end.

Author Response: Due to the large number of figures we were not able to place all of them inline. We anticipate that all figures will be appropriately positioned in the final publication, and will work with the editors to ensure that is the case.

Reviewer 3

Reviewer 3-1 In the Participants section the text would benefit from the presence of the demographic table, now included in the supplementary material, with the related statistical comparisons . It would also be useful to include the sub-sample used for the validation set within this table, with related comparisons with the total group, in order to support the text in the results section (pg 10).

Author Response: Thank you for this great suggestion. The table with demographic information that was included within the Supplementary materials has now been moved into the main body of the manuscript, in the Participants section on page 6. Separate columns were added to this table to report demographic information just for the sub-sample included in the validation experiments. We now also refer to this information in the Results section where we report results comparing the validation sample with the total sample.

Reviewer 3-2 Data Collection setup: It is not clear from the text which subjects have been assessed with the ESCS scale and which ones with the BOSCC and which criteria have been chosen to carry out the assessment. Instead, it seems to me that for the ADOS scale all subjects with a suspicious autism spectrum disorder have carried out the evaluation. These information should be reported in the text or in table. Could the authors provide it?

Author Response: We agree that the description of the assessment protocols needed further clarification. In short, all study participants completed both the ESCS and the BOSCC play interactions and were used for training and validating the eye contact detector.

For the BOSCC scores, only 25 minimally-verbal subjects within ASD group were given scores as they are considered appropriate for scoring per the BOSCC manual.

In order to clarify this process, we created a new diagram presented in Figure B, and updated the text in the Participants section and Data collection setup section for clarification. This diagram has been added in the manuscript as Figure 3.

Reviewer 3-3a Page 10 see point 1. Please provide Mean and STD of the statistical comparison cited in the text.

Author Response: The Means and Standard Deviations for the t-tests reported in this paragraph are now included in the text.

Reviewer 3-3b Effect size are missing in the analysis , please the author provide it.

Figure B: Study subjects breakdown.

Author Response: We conducted two sets of hypothesis tests in our study. The first is to test the equivalence between human coders and deep model, and the second is to re-confirm findings of prior studies.

In the equivalence test, information on the effect size has been added in the Reliability with human raters section under Results (on page 14). In the reproducibility analysis, we now computed the effect sizes for all of the statistically significant findings and included them in the respective tables (Table 5 and 6).

Reviewer 3-3c Did you look at gender differences in the data?

Author Response: Thank you for this great suggestion. To examine if gender affects the performance of eye contact detection, we divided our validation sample by gender and repeated the precision/recall analysis, illustrated in Figure C. As can be seen, the curves appear similar across the two groups. In both cases, the model’s output (red diamond) is within one SD (green

error bar) of the average of human experts, indicating an absence of gender effects. This additional analysis has been added to the Supplementary Information.

Figure C: Precision and recall split by gender.

Reviewer 3-3d Correlational Analysis. Would be useful show if also the frequency and duration of eye contact, coded by human raters are correlated with the clinical assessment. In general from a clinical point of view would be interesting see if this automated coding is able to show differences in the frequency and duration of eye contact able to show a difference between typically children compared with the ASD group, in order to find potential biomarkers.

Author Response: We realize that the description in the paper was not sufficiently clear regarding the fact that not all subjects have been coded for eye contact using their entire video data. In our dataset, eye contact was annotated for both BOSCC and ESCS assessments for 45 subjects. For the remaining 76 subjects, eye contact annotations were only completed for one of the assessments (n=47 for the ESCS; n=29 for the BOSCC) due to the time-consuming nature of the manual coding process. This information is now included in the text under Materials and Methods / Video coding of eye contact, page 7.

Given this limited availability of human coding, we are not able to re-run the correlation analysis using the same set of samples. For example, the reported correlation of BOSCC SA \times Eye Contact uses $N = 25$, but only 17 of them have human annotations for eye contact during the BOSCC recording. Nevertheless, we were still able to observe that manually coded eye contact and severity scores show equally strong correlations ($N = 17$. frequency: $r = -.75$, $p < .001$. du-

ration: $r = -.77$, $p < .001$), compared to those based on automated coding ($N = 25$. frequency: $r = -.75$, $p < .001$; duration: $r = -.78$. $p < .001$).

In addition, based on the reviewer's suggestion, we performed the precision/recall analysis separately for the ASD and TD group to determine if diagnostic status affects the performance. We found that the model's performance is still within one SD of the average of the human raters, indicating equivalence (Fig. D). This figure has been added to the Supplementary materials. Achieving human-level performance across the diagnostic groups is a key step towards finding potential biomarkers of ASD using automated measures, and we hope our work facilitates interest in this direction in future research efforts.

Figure D: Precision and recall split by diagnostic category.

Reviewers' Comments:

Reviewer #1:

Remarks to the Author:

Regarding the following author's reponse, please include the result of the 'additional experiments' in the main text (I could not find in the updated manuscript to describe this part) including the figure describing the relation of the distance vs. Eye contact F1 score which is important information for readers.

This is because the 'unaffected minimum distance' (95cm) is not too short in children-parent communications. Rether, shorter distance is quite important for communications.

> We conducted an additional simulatio analysis by down-sampling the face images that are input to our model, to capture the presentation of faces over a range of distances. We found that the accuracy of our model is unaffected by interpersonal distances of up to 95cm (Eye contact F1 score: model = 0.940, average human = 0.932). Moreover, performance is minimally effected by distances of up to 2.3 meters (F1 score: model = 0.931). This distance range is more than adequate to cover scenarios of face-to-face interaction. For example, this range covers the zones of intimate and personal space defined in the study of proxemics (Hall. 1963) in which person-to-person social interactions occur.

Reviewer #2:

Remarks to the Author:

The revised version of the paper is much better written, and seems to have accounted for most concerns from me as well as fellow reviewers. There are still some typographical and grammatical errors in the manuscript, and I hope the authors can carefully proofread the paper before re-submitting it.

Overall, I am in favor of accepting the paper.

Point-by-Point Response to Referees' Comments

Reviewer 1 Regarding the following author's response, please include the result of the 'additional experiments' in the main text (I could not find in the updated manuscript to describe this part) including the figure describing the relation of the distance vs. Eye contact F1 score which is important information for readers. This is because the 'unaffected minimum distance' (95cm) is not too short in children-parent communications. Rather, shorter distance is quite important for communications.

Author Response: We have now included the figure and description for this experiment in the Supplementary Information.

Reviewer 2 The revised version of the paper is much better written, and seems to have accounted for most concerns from me as well as fellow reviewers. There are still some typographical and grammatical errors in the manuscript, and I hope the authors can carefully proofread the paper before re-submitting it. Overall, I am in favor of accepting the paper.

Author Response: We carefully reviewed the final version of the manuscript to ensure that there are no typographical and grammatical errors.